

# GIS-based Real-time Framework of Debris Flow Hazard Assessment for Expressways in Korea

C. K. Chung[1], H. S. Kim[2], S. R. Kim[2], K. S. Kim[3]

[1]Department of Civil and Environmental Engineering, Seoul National University, Seoul, Korea
[2]Earthquake Research Center, Korea Institute of Geoscience and Mineral Resources, Daejeon, Korea
[3]Expressway & Transportation Research Institute, Korea Expressway Corporation, Hwaseong, Korea

*Correspondence to*: H. S. Kim (adoogen@kigam.re.kr)

**Abstract.** Debris flows caused by heavy rainfall in mountain areas near expressways lead to severe social and economic loss and sometimes even result in casualties. However, in Korea, the design of road structures that resist these debris flow incidents are generally not carried out in a systematic way with proper concepts or procedures. Therefore, the development of a real-time system for debris flow hazard assessment is necessary to provide preliminary information for rapid decision making of evacuations or restoration measures, and to prevent second-hand disasters caused by debris flows. Recently, various map-based approaches have been proposed using multi-attribute criteria and assessment methods for debris flow susceptibilities. However, for the macro-zonation of debris flow hazards at a national scale, a simplified method such as the Korea Expressway Corporation debris flow hazard assessment method is appropriate and also applicable for systemization based on GIS and monitoring networks. In this study, a GIS-based real-time framework of debris flow hazards for expressway sections was newly proposed based on the KEC debris flow hazard assessment method. First, the KEC-based method was standardized in a systematic fashion using ESRI ArcGIS, enabling the objective and quantitative acquisition of various attribute datasets. Also, for a more precise assessment, the quantification of rainfall criteria was considered. Finally, a safety management system for debris flow hazards was developed based on a GIS platform, and was applied and verified on three expressway sections in Korea.

## 1 Introduction

Landslides, and in particular, debris flows are one of the most damaging natural hazards in mountainous terrains with heavy torrential rainfall as is in the case in Korea. Rainfall-triggered landslides are a recurring problem in Korea due to the mountainous terrain with a shallow layer of alluvial soil, and associated weather conditions (Park et al., 2013). 'Flowage', or 'flow', is the term used to describe the downslope movement of unconsolidated material in which particles move about and mix within the mass, such as earthflows, debris flows, or avalanches. To avoid confusion resulting from the interchangeable use of terminology, the term 'debris flow' in this study refers to the fluid mixture of rocks, sand, mud, and water that is intermediate between a landslide and a water flood (Keller and DeVecchio, 2012; Feusto et al., 1999).



Debris flow damage includes loss of human life, destruction of various facilities, and damages to roads, pipelines, and vehicles (Jakob and Hungr, 2005). Until now, only post-event repair processes and works have mostly been executed after debris-flow occurrences. Recently, an increase has occurred in both the number of occurrences and costs for countermeasures of debris-flows in Korea. In order to sufficiently manage expressway sections and facilities from debris

flow occurrences, a method to assess the hazard of debris flows during certain rainfall events at a regional scale is needed. Infiltration of prolonged intense rainfall, causing soil saturation and a temporary increase in pore water pressure, is the mechanism by which most shallow landslides, and more specifically debris flows, are generated during rainstorms (Iverson, 2000; Keller and DeVecchio, 2012). Because Korea is a region with heavy concentrated torrential rainfall in the summer, most of the debris flows occur during the rainy summer periods in mountainous regions. Due to global warming and various

environmental factors, the intensity and frequency of rainfall events are increasing, resulting in a greater number of debris flow occurrences and restoration costs than before. Consequently, more debris flow countermeasures are now necessary for the safety and preservation of human lives and infrastructures.

Assessment of landslides including debris flows has been carried out with GIS techniques combined with statistical analyses and physical-based approaches by various researchers such as Dai et al. (2002), Ohlmacher and Davis (2003),

Ayalew and Yamagishi (2005), Wang et al. (2008), and Kritikos and Davies (2014). Through comparison and review of existing studies on debris flow influential factors and assessment methods, the Korea Expressway Corporation (KEC) debris flow hazard assessment method (Expressway & Transportation Research Institute, Korea Expressway Corporation, 2009), which focuses on the likelihood of road structure hazards, was developed as a fundamental assessment tool. The method can be quantitatively and objectively performed in a simple manner by using documents such as numerical maps and expressway

design files, minimizing the need for tiresome field investigations in countless potential debris flow occurrence regions in vast areas. Above all, in order to immediately assess rainfall-induced debris flows, rainfall criteria were utilized for fundamental trigger values and indexes for severity levels of debris flows.

According to the KEC method, debris flows are evaluated through two indexes: the *Susceptibility Value* and the *Vulnerability Value*. The *Susceptibility Value* indicates the likeliness of whether a debris flow will occur, and can be

estimated with the topography information on target locations. The *Vulnerability Value* is used to represent whether an occurred debris flow will actually damage or have an impact on certain expressway sections, and can be assessed from the capacity of the drainage facility and the margin area for sedimentation of debris flow materials before they reach expressway structures. Determining these influential factors is achieved through the use of digital maps and expressway design files. The calculated *Susceptibility Value* and *Vulnerability Value* are used to indicate a single *Hazard Class* varying from S to E,

which represents the likelihood of damage by debris flow events in a given rainfall intensity.

In this study, the GIS-based real-time framework of debris flow hazards for expressway sections was newly proposed based on the KEC debris flow hazard assessment method. First, to standardize the KEC method, a systematic sequence for the acquisition of attribute values was newly proposed using an objective tool, ArcGIS. Second, considering the real-time link with the Automatic Weather Station (AWS) network of Korea, the rainfall reoccurrence periods were quantified as



accumulated rainfall using the trigger values of past debris flow occurrence events. Following the optimization of the KEC debris flow hazard assessment method, a safety management system for debris flow hazards was developed based on the aforementioned GIS platform. This system consists of a database (DB) and three systematic sub-modules; the input module, the debris flow hazard assessment module, and the real-time debris flow hazard assessment module linked to the rainfall

5   monitoring network. Finally, an applicability evaluation for the framework was carried out on three expressway sections of Korea, that have suffered damage due to debris flow occurrences during heavy rainfall events in 2005 and 2006: the Pyeongchang area of the Yeongdong Expressway, the Deogyu Mountain area of the Daejun-Jinju Expressway, and the Juksan-Geochang area of the 88 Expressway. The reliability of the assessment method was investigated by comparing the actual debris-flow occurrence and non-occurrence cases.

## 2 Literature review of debris flow hazard assessment

Through the review of methods for debris flow hazard assessment including those by Dai et al. (2002), Lin et al. (2002), Ayalew et al. (2004), Lee and Pradhan (2007), Blahut et al. (2010), and Lee et al. (2012), the influential attributes and hazard assessment methods of debris flows were comparatively analyzed. Debris flow hazards are dependent on a specific set of factors and processes that are usually investigated by various experts (such as hydrologists, geologists, or civil engineers).

Prediction concerns either where or when debris flow will occur, depending on the type of movement and the scope of the forecasting. Many attributes (geography, rainfall, geology, vegetation, wildfire history, and conditions of existing structures) are related to the mechanism/initiation of debris flows (Table 1).

To obtain information on the influencing factors other than topographical properties (elevation, slope, valley and watershed) and rainfall data, field surveys should be thoroughly conducted throughout entire expressway facility sections.

Because the assessment of debris flow hazards in this study is to be applied on a regional scale, the method needed to be simple, and also applicable for the macro-zonation of debris flow hazards. Considering that obtaining and processing all the attributes stated above in the prediction stage is a time-consuming and difficult task, it was decided that only easily accessible document data (such as digital maps, geological maps, etc.) were to be used in the assessment process. Consequently, the KEC debris flow hazard assessment method was set as a fundamental assessment tool owing to its

simplicity.

As a precautionary measure, the KEC method assesses the hazard of debris flows at a regional scale using a limited number of data sets. Only Digital Elevation Models (DEMs) and expressway design files of the target area were used. The DEMs that were used in the assessment process were those provided by the National Geographic Information Institute (NGII) of Korea, whereas the expressway design files were provided by the Korea Expressway Corporation. Through the

application and use of only easily accessible datasets, the KEC hazard assessment method minimizes the need for tedious and time-consuming field investigations, allowing an easy and comfortable hazard assessment of debris flows in a large area. The debris flow hazard is evaluated through two indexes: the *Susceptibility Value* and *Vulnerability Value*.



The *Susceptibility Value* indicates the likelihood of whether debris flow will occur in a target area and is assessed using a total of four attributes. The mean watershed slope, and area percentage of watershed with slopes over 35° are used for the assessment of debris flow initiation. The mean valley slope, and length percentage of valley with slopes over 15° are used for the assessment of debris flow movement. Other factors such as the size and shape of the valley along with the variations in

slope direction, properties of the subsoil, and vegetation also have an influence on the initiation and movement of debris flows. However, to simplify the method, only the slope information derived from the DEMs was considered for debris flow possibilities. Each of the influential factors are given points from 0 to 5 based on the grading standard set by past debris flow occurrence cases (Table 2), and adds up to a total *Susceptibility Value* of 20 points. For the weight considerations of the four attributes, logistic regression was carried out through the Statistical Package for Social Science (SPSS). Results showed that

the 4 *Susceptibility Value* attributes had weights of 0.27, 0.24, 0.26, and 0.23, respectively. Since the weights showed no significant difference, the attributes were considered to have identical weights.

The *Vulnerability Value* indicates whether a debris flow will actually damage or have an impact on expressway sections. The *Vulnerability Value* is assessed by two attributes: the volume of margin area to deposit debris flow materials before reaching expressway structures, and the size of drainage facilities running through the expressway. For the acquisition of the

attribute values, expressway design files provided by the Korea Expressway Corporation were used. Each of the attributes is given points ranging from 0 to 5 based on a grading standard (Table 2), and these points are added up to provide the total *Vulnerability Value* of 10 points.

With the integrated *Hazard Value* using calculated *Susceptibility Value* and *Vulnerability Value*, a *Hazard Class* is given for a target expressway section. In the table of severity rating shown in Fig. 11, the x-axis and y-axis indicate the

*Vulnerability Value* and *Susceptibility Value*, respectively. Through investigations on past debris flow occurrences, the *Hazard Classes* were categorized according to the rainfall reoccurrence period for expressway design purposes. *Hazard Class* S indicates a likelihood of debris flow occurrences in areas with rainfall reoccurrence periods of 2 to 5 years. *Hazard Classes* A, B, C, and D have rainfall reoccurrence periods of 5 to 20 years, 20 to 50 years, 50 to 100 years, and over 100 years, respectively. *Hazard class* E indicates an area with a very low likelihood of debris flow damage (Expressway and

Transportation Research Institute, 2009).

## 3 GIS-based framework for debris flow assessment

For the processing of attributes included in the KEC method, a systematic sequence using the ArcGIS 10.1 software was newly proposed (Fig. 1). Various ArcGIS tools such as the [Spatial Analyst Tools] and the [Analysis Tools] were used for a quantitative and objective assessment of the attributes.



## 3.1 Attribute Processing for the *Susceptibility Value*

For the processing of watershed slope and valley slope datasets, DEMs provided by the NGII of Korea were used. Numerical maps with the highest resolution were those of a scale of 1:1,000. However, 1:1,000 scale numerical maps were only provided for major urban areas. Because numerical maps of the highest resolution provided for the entire Korean Peninsula were those of 1:5,000 scale, numerical maps with the scale of 1:5,000 were implemented in the attribute processing for the *Susceptibility Value*.

Of the entities within the DEMs, only the polyline entities having no elevation value were extracted from the numerical map and used for construction of DEM. Because the system focuses on the debris flow hazard assessment of expressway facilities, the expressway layers were selected. For the processing of slopes in the surround area of the expressway, the elevation layers were also selected (Fig. 2a). With the elevation layers of DEMs, the elevation and slope raster with the smallest cell sizes possible were obtained, as shown in Figs. 2b and 2c. Because the minimum cell size that could be considered with 1:5,000 DEMs was 5 meters, the raster with cell sizes of 5 by 5 meters were processed. Based on the elevation raster, the flow direction data sets were computed. The [Flow Direction] tool creates a raster of flow direction from each cell to its steepest downslope neighbour (Olivera et al., 2002). From the flow direction raster, the flow accumulation datasets were obtained (Fig. 2d). The [Flow Accumulation] tool creates a raster of accumulated flow into each cell. With a flow accumulation grid, valleys can be defined through the use of the flow accumulation value (Olivera et al., 2002) (Fig. 2e). For a more accurate visualization of valley areas, the properties of the flow accumulation grids were altered in various ways. Through trial and error, along with comparison with the actual field investigations, the final standard deviation of 0.1 was used to visualize the valleys in the most appropriate and realistic way.

After setting a pour point (output point) on the route of the assessed expressway, the flow direction and pour point were taken into consideration to obtain the watershed. The [Watershed] indicates the drainage areas contributing to the flow from the land surface to the water system (ESRI, 2002) (Fig. 2f). Through the [Extract by Mask] tool, the slopes of the cell in the watershed area were obtained. Through the histogram in the raster properties, the values for the attributes of the mean watershed slope and area percentage of the watershed with slopes over 35° were acquired. Because the slope raster indicates the steepest slope with regard to the surrounding pixels of the flow direction, and not the slope in the valley direction, the previously obtained slope raster could not be implemented for the attributes related to the valley. Other means should be used for the acquisition of the valley slope data. An approach that assesses the slope through the elevation raster in the valley direction was proposed and applied.

From the flow accumulation layer, the valley shapes in the watershed were obtained. The valley paths were manually plotted on the elevation layer. The elevations of the cells in the path of the valley were obtained through the [Extract by Mask] tool using the plotted valley path and elevation layer. With the length of the plotted valley path, and the total elevation difference between the expressway and the highest point in the valley path, the mean valley slope was simply calculated as follows:





$$\text{Mean valley slope } (\Phi_i) = \tan^{-1}\left(\frac{\Delta H}{L_{path}}\right) \tag{1}$$

$\Delta H$ is the elevation difference between the expressway and the highest point in valley path, while $L_{path}$ is the length of the

plotted valley.

Considering the fact that the assessment method will be implemented on a real-time hazard assessment method, the

method needed to properly assess debris flow hazards in a precautionary fashion. In the estimation and prediction stage, the

valley in which a debris flow will initiate is unknown. Therefore, for watersheds with more than one valley, all valleys in the

watershed were considered in the process. The mean valley slopes for each valley were calculated, and were averaged using

the valley lengths as weight factors:

$$\text{Overall mean valley slope} = \frac{\sum_{i=1}^{n}(\Phi_i \cdot L_i)}{\sum_{i=1}^{n} L_i} \tag{2}$$

$\Phi_i$ is the mean valley slope of the individual valley and $L_i$ is the length of the individual valley separated by the watershed

DEMs.

The extracted DEMs obtained through the [Extract by Mask] tool did not properly represent the valley directions,

showing abrupt valley direction changes at right angles (Fig. 3a). In order to appropriately assess the slopes in the actual

flow directions of the valleys for the calculation of the length percentage of the valley with slopes over 15°, additional

ArcGIS tools were used. The extracted valley path elevations of cells were converted to points through the [Raster to Point]

tool, and were assigned to the map coordinate system (Fig. 3b). With the [Buffer] tool, a buffer of 1.5 meters was set around

the manually plotted valley path (Fig. 3e). In order to obtain the valley path points that are positioned inside the 1.5m buffer

zone of the plotted valley path, the [Intersect] tool was used. Through the process, only the valley path points that were in the

vicinity of the actual valley path were obtained (Fig. 3c). Using the dbf files of extracted points, the slope between the DEMs

in the direction of the valley travel path was calculated using Microsoft EXCEL. Through the calculation results of the type

of dbf files, the distances between cells with slopes of over 15° were obtained, allowing the calculation of the length

percentage of a valley with slopes over 15°. According to the mean valley slope calculation process in watersheds with more

than one valley, the length percentage of valley with slopes over 15° was calculated using a similar process:

$$\text{Overall length percentage of valley with slopes over } 15° = \frac{\sum_{i=1}^{n}\{(L_{15})_i\}}{\sum_{i=1}^{n} L_i} \tag{3}$$

$(L_{15})_i$ is the length of valley with slopes over 15° for the individual valley.

### 3.2 Attribute Processing for the *Vulnerability Value*

From the expressway design files, the volume of area available for sedimentation was calculated by simplifying the area as a

triangular pyramid bounded by the valley and expressway embankment (Expressway and Transportation Research Institute,

2009) (Fig.4). The volume of the margin area of sedimentation is calculated using Eq. 4. The drainage facility information of

each watershed was obtained through the expressway design files, and the attribute points were determined according to the

grading standard set by the KEC (Table 2).



$$\text{Sedimentation Volume (m}^3) = \frac{1}{6}(L \times H \times W) \tag{4}$$

### 3.3 Quantification of rainfall criteria for *Hazard Class*

According to the KEC method, the rainfall reoccurrence periods (ranging from 2 years to more than 500 years) were used as trigger values of the landslide hazard. However, to immediately determine the rainfall event by linking the network sever of meteorological observatories and real-time estimates of the landslide hazard, it is impossible to directly apply the rainfall reoccurrence periods for the proposed framework. Thus, the rainfall reoccurrence periods were quantified as representative accumulated rainfall criteria: 1-hour rainfall (Yun et al. 2010), 6-hour rainfall (Ham et al. 2014, Oh and Park 2013), and 3-day rainfall (Yoo et al. 2012). The baseline rainfalls were set at the lower bounds of 1-hour, 6-hour, and 3-day rainfall data from AWS and at the major interchanges at eighteen stations on the 9 expressway sections in which the highest number of debris flow events occurred in Korea. Also, various sets of these rainfall baselines using accumulated rainfall data recorded from AWS were cross-validated with the actual damage state for early warning. These referred to the recurrence intervals of rainfall corresponding to the *Hazard Class* in the KEC-based debris flow assessment (Choi et al. 2015). From the correlations between reoccurrence period and accumulated rainfall data regarding to the *Hazard Class*, the range and lower bound of 1-hour,6-hour and 3-day accumulated rainfall datasets were determined (Table 3). Thus, the rainfall criteria were set with the lower bound values of accumulated rainfall ranges for five *Hazard Classes* based on conservative hazard management.

The severity levels of debris flow were defined as three levels: safe, caution, and warning. If the measured rainfall datasets do not correspond to all three cases of accumulated rainfall datasets, the target express section is considered as safe from debris flow hazard. However, if more than one indicator exceeds the three baseline rainfalls upon the five *Hazard Classes*, the debris flow hazard needs to be forecast as a level of caution or warning. The criteria are appropriate for conservative risk management because of the cross validation between rainfall datasets and *Hazard Class* in the case of actual damage status on the representative 9 expressway sections. Through the quantified rainfall criteria for the *Hazard Class* of debris flow, although it is desirable to apply the actual probability rainfall of the possible reoccurrence period (Choi et al. 2015), difficulties occur in the intensive management of debris flow hazard for the national expressway. Therefore, an additional simulation for further locations is required to establish the appropriate rainfall criteria based on the debris flow risk assessment.

## 4 Safety management system for debris flow hazard

The GIS-based safety management system for debris flow hazard consists of a database (DB) and systematic modules (Fig. 5). The database contains all field data and processed data in the system. The sub-modules execute various functions on managing and utilizing information in the database. These include three systematic sub-modules; the input module, the





debris flow hazard assessment module, and the real-time debris flow hazard assessment module linked to the rainfall monitoring network. The framework including all these functions focuses on user-friendliness and real-time applications.

### 4.1 System database

DB is the backbone of the developed framework. It stores not only primary collected data such as the digital numerical map, expressway, and real-time based rainfall monitoring data (which is standardized using Automatic Weather Station (AWS), transmitted from meteorological observatory server) but also secondary processed data obtained from the debris flow hazard assessment and real-time prediction (Fig. 6). It contains all data as alphanumeric values according to standard data formats, which are the outcome of data classification and standardization with spatial information. The data stored in the database can be easily utilized in the framework. The digital numerical map can be used as the basic topographical information of the system because it offers an easy way to construct topographical information for a target area. Also, the expressway information is composed of organization categories for managing Korean expressway route data, and spatial datasets designed for various coordinates systems (longitude and latitude, GRS80, Google coordinates). The data format of rainfall events (belonging to AWS) was arranged previously for an expressway in Korea. In rainfall monitoring based on real-time, rainfall observatory data, and rainfall monitoring data were standardized.

### 4.2 System sub-modules

Input function provides an effective way to store and arrange all collected field data including electric or non-electric documents, general information, digital numerical map, expressway data, rainfall monitoring data, and analysis data, according to a standardized data format based on DB. Especially, based on the rainfall criteria of severity levels for debris flow, the target expressway section are grouped according to influence of the spatial range (having 5km radius), under the nearest AWS in Korea (Fig. 7). The reclassified AWS datasets for rainfall criteria (1-hour, 6-hour, and 3-day accumulated rainfall datasets) are linked to the grouped expressway sections using the system database.

The real-time debris flow framework has four functional phases with the database based on the proposed schematic sequence of debris flow hazard assessment (Fig. 8). In the first phase, linked with the digital numerical map and DEM, the watershed DEM, and valley layer are extracted using the ArcGIS desktop program and input system DB. In the second phase, the *Susceptibility Value* and *Venerability Value* for the target route are constructed into DB combined with geospatial information. In the third phase, to transmit the reliable rainfall monitoring data for the target route from the widely distributed AWS server on a real-time basis, the routes of the completed site investigation for debris flow hazards were grouped into the same datasets focusing on the adjacent rainfall station in certain areas. In addition, the rainfall values for debris flow hazard assessment are automatically computed based on the monitoring criteria with rainfall recurrence periods for road design in Korea when the input rainfall monitoring data is input to DB. Finally, following the rainfall threshold level for debris flow hazard (from the KEC method), the severity levels according to safe, caution, and warning are determined



using map symbols (blue, yellow, and red symbols, respectively, in Fig. 8) at the target route in real-time. In addition, the sound signal and message window are alarmed to notify the expressway administrator of the hazard status.

The debris flow hazard prediction function shows all attributive information in the database by using tables and graphics according to its characteristics either, on screen or as a document. Also, all data in the DB can be output as a chart or graph. The graphic functions simultaneously display interpolated data with field data over an arbitrary domain. All of the charts, graphs, and drawings can then be printed. Especially, the debris flow hazard can be visualized and forecast as 2D maps overlaid with satellite images. Also, the severity level of the target route can be determined using zonation criteria in real-time.

In this proposed framework, the computer-based method for the real-time assessment of spatial debris flow hazard was embedded based on a stand-alone system developed using Microsoft Visual BASIC, the Esri ArcGIS developer tool (Esri, 2006; Lee and Wong, 2001). The ArcGIS developer tool was mainly used for the development of the database, evaluation of the results, and spatial visualization. Several assumptions and precedent assessments are needed to estimate the possible debris flow hazard for a target site in real-time, at the point at which the debris flow occurs. Especially, the precedent procedures consist of the building of the database, DEM construction, and determination of *Susceptibility* and *Venerability value* in order to consider the site-specific debris flow potential overall target area, prior to the occurrence of debris flow and rainfall. As the debris flow occurs near the target site, the possible severity level can be estimated in real time by linking with the rainfall data monitored from the AWS server.

## 5 Application of assessment method on selected sites

### 5.1 Condition of application

The Pyeongchang area of the Yeongdong Expressway, the Deogyu Mountain area of the Daejeon-Jinju Expressway, and the Juksan-Geochang area of the 88 Expressway in Korea were selected for investigation of their debris flow hazards (Fig. 9). Many debris flows occurred in the Pyeongchang area of the Yeongdong Expressway during a heavy rainfall event (244.0mm/day, 66.0mm/hr) in the summer of 2006 (Table 4). Debris flows occurred in the Deogyu Mountain area of the Daejeon-Jinju Expressway in the summer of 2005. The rainfall intensity in the region at that time was 312.0mm/day and 54.5mm/hr. The Juksan-Geochang area faced several debris flows during a heavy rainfall event in the summer of 2006. The rainfall intensity at that time was 121.0mm/day, and 31.5mm/hr. When comparing all target areas, the Pyeongchang and Deogyu Mountain areas had similar rainfall intensities, indicating the same rainfall reoccurrence periods, whereas the Juksan-Geochang area had the lowest daily rainfall. For each area, a test bed area was set for the application of the assessment method. A test bed with a length of 11km along the Yeongdong Expressway was chosen for the Pyeongchang area. The lengths of the test beds for the Daejeon-Jinju Expressway and 88 Expressway were 3km and 2km respectively, presented as black dotted lines in Fig. 9.



## 5.2 Verification of framework for debris flow assessment

All existing watersheds in the selected expressway test beds were analyzed. Of all the watersheds in the selected regions, the areas with expressways positioned on bridges and tunnels, or near vast areas of fields were excluded from the analysis due to their very low likelihood of damage. Since reported debris flows were based on the damage made to road structures, those areas without any damage were not reported. Thus, no debris flow damages were reported in regions where the volumes of possible sedimentation are vast, and consequently were not considered in the analysis process. After the exclusion of the aforementioned sites, the watersheds were then classified according to whether or not debris flow damages were reported. Areas with debris flow damages reported are hereinafter referred to as "occurrences", and those without damage reports are referred to as "non-occurrences". As a result, 18 debris flow occurrences and 14 non-occurrences were analyzed for the Pyeongchang area based on the proposed framework for debris flow assessment. Twelve debris flow occurrences and 8 non-occurrences were analyzed for the Deogyu Mountain area. Nine debris flow occurrences and 7 non-occurrences were analyzed for the Juksan-Geochang area as shown in Fig. 10.

Applications of the KEC method show results which roughly coincide with the actual debris flow occurrences and non-occurrences shown in the table of *Hazard Value* (varying from 0 to 30) and *Hazard Class* (Fig. 11). Occurrence cases are roughly positioned in the upper right-hand side, which indicate higher *Susceptibility* and *Vulnerability Values*, whereas non-occurrence cases are located on the lower left side, with relatively lower *Susceptibility* and *Vulnerability Values*. Although this tendency may seem correct to some extent, it does not always show flawless results. In the *Hazard Classes* of C and D, both debris flow occurrence and non-occurrence cases are mixed up, not always indicating a result in which occurrences have higher *Hazard Classes*, and non-occurrences with lower classes. Debris-flows occurred even in areas with a *Hazard class* of E, which was initially intended to indicate a very low likelihood of debris-flow. The total sum of *Susceptibility* and *Vulnerability Values* even showed a greater number of non-occurrences than the occurrences due to the low *Hazard Values* of occurrence cases.

Although the sites showed results that generally presented higher *Susceptibility Values* and *Vulnerability Values* in occurrence cases, they did not perfectly represent the total *Hazard Value* differences between occurrences and non-occurrences. However, the effectiveness of the proposed framework was validated for three cases because the potential of debris flow hazard is highly evaluated for the actual occurrence cases. In addition, the average value of the *Susceptibility Value*, *Vulnerability Value*, and *Hazard Value* of occurrence cases are 1.15 times greater than those of non-occurrence cases (Table 5). In order to appropriately represent the differences between occurrences and non-occurrences of debris flows, modifications need to be made on the grading standard of the KEC method from a number of various validation tests based on the proposed framework. Specifically, grading standards for attributes need to be revised to more well-founded standards that take into consideration the attribute values of both occurrences and non-occurrences, maximizing the distinction between the two instances. Also, attributes other than those regarding the slope should be considered such as watershed size and bending of valley (Kim et al. 2014).



## 6 Conclusions

In this study, to provide preliminary information for rapid decision making of evacuations or restoration measures, and to prevent second-hand disasters caused by debris flows, a GIS-based real-time framework of debris flow hazards for expressway sections was newly proposed based on the KEC debris flow hazard assessment method. First, the KEC-based method was standardized using ESRI ArcGIS, enabling various attribute datasets to be acquired in an objective and quantitative manner with a fixed data acquisition sequence. Second, considering the real-time link with the AWS network, the rainfall reoccurrence periods were quantified as accumulated rainfall using trigger values of actual debris flow events.

Finally, based on the optimized KEC debris flow hazard assessment method, a safety management system for debris flow hazards was developed based on a GIS platform, and then applied on three expressway sections in Korea. The GIS-based safety management system for debris flow hazards consists of a DB and functional sub-modules. The DB contains all field data and processed data in the system. The sub-modules execute various functions on managing and utilizing information in the database. Three sub-modules are used the input module, the debris-flow hazard assessment module, and the real-time prediction module of debris flow hazard. The framework including all these functions focuses on user-friendliness and real-time applications. To estimate the possible debris flow hazard in real-time, several assumptions and preceding assessments were made such as the database establishment, DEM construction and determination of *Susceptibility* and *Vulnerability Value* standards. As the predetermined *Hazard Values* are combined with rainfall data in real-time, the likelihood or severity level of debris flows can be estimated in real-time. Immediately after assessment of the danger levels are assessed, a set of caution or warning levels are initiated, allowing proper countermeasures to be rapidly carried out for target sections at risk.

For field applicability evaluation, the framework was applied on three expressway sections in Korea: the Pyeongchang area of the Yeongdong Expressway, the Deogyu Mountain area of the Daejeon-Jinju Expressway, and the Juksan-Geochang area of the 88 Expressway. The reliability of the assessment method was investigated by comparing actual debris-flow occurrence and non-occurrence cases. The method, however, did not perfectly fit the actual occurrences and non-occurrences due to the limitation of the data considered in the process. In order to properly assess the debris-flow occurrence hazards, more reliable rainfall criteria should be considered, and modifications should be made on the grading standard for each individual influential attribute considered in the KEC method. Also, additional attributes should be considered in the hazard assessment such as watershed size and bending of valley.

*Acknowledgements.*



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



**Table 1.** Current debris flow influence factors and survey methods.

| Influence factor | | Survey method |
|---|---|---|
| Topographical property | - Elevation<br>- Slope (angle, direction and shape)<br>- Valley (length, slope and width)<br>- Watershed (area and slope) | - Digital map<br>- DEM (Digital Elevation Model) |
| | - Initial zone (area, volume, shape and failure type) | - Field survey |
| Hydrological property | - Rainfall (Max., intensity, reoccurrence period and accumulated rainfall) | - AWS (Automatic Weather Station) |
| Geological property | - Lithology<br>- Specific angle<br>- Moisture content<br>- Void ratio<br>- Porosity<br>- Saturation<br>- Density<br>- Permeability<br>- Ground water | - Soil map<br>- Field survey |
| Vegetation property | - Tree (species and tensile strength on the root)<br>- Wildfire history | - Stock map<br>- Field survey |
| Structural fragility | - Sedimentation volume<br>- Size of drainage<br>- Fragility curve | - Design file |





**Table 2.** Points given to attributes according to grading standard of KEC (Expressway and Transportation Research Institute, Korea Expressway Corporation, 2009).

| Classification | | | Scoring Criteria | |
| --- | --- | --- | --- | --- |
| | | | Scoring Index | Points |
| *Susceptibility Value* (20 Points) | Initiation Assessment (10 Points) | MeanWatershed Slope (Unit : °) | - Higher than 35° | 5 |
| | | | - 30°~35° | 4 |
| | | | - 25°~30° | 3 |
| | | | - 20°~25° | 2 |
| | | | - 15°~20° | 1 |
| | | | - Under 15° | 0 |
| | | Area Percentage of Watershed with Slopes over 35° (Unit : %) | - Higher than 40% | 5 |
| | | | - 30%~40% | 4 |
| | | | - 20%~30% | 3 |
| | | | - 10%~30% | 2 |
| | | | - 1%~10% | 1 |
| | | | - Under 1% | 0 |
| | Movement Assessment (10 Points) | Mean Valley Slope (Unit : °) | - Higher than 25° | 5 |
| | | | - 20°~25° | 4 |
| | | | - 15°~20° | 3 |
| | | | - 10°~15° | 2 |
| | | | - 5°~10° | 1 |
| | | | - Under 5° | 0 |
| | | Length Percentage of Valley with Slopes over 15° (Unit : %) | - Higher than 90% | 5 |
| | | | - 70%~90% | 4 |
| | | | - 50%~70% | 3 |
| | | | - 30%~50% | 2 |
| | | | - 10%~30% | 1 |
| | | | - Under 10% | 0 |
| *Vulnerability Value* (10 Points) | Debris Storage, Sedimentation Availability (5 Points) | Volume of Deposit Area (Unit : $m^3$) | - No accumulation area ($0m^3$) | 5 |
| | | | - $0m^3$~$100m^3$ | 4 |
| | | | - $100m^3$~$1,000m^3$ | 3 |
| | | | - $1,000m^3$~$5,000m^3$ | 2 |
| | | | - Higher than $5,000m^3$ | 1 |
| | | | - Excessive volume of deposit area, No damage guaranteed | 0 |





| Debris Passage through Expressway Facilites (5 Points) | Size of Drainage Facility (Unit : Cross-sectional Area, $m^2$) | - Waterway | 5 |
|---|---|---|---|
| | | - Lateral drains below D1,200 | 4 |
| | | - Waterway box below B2.0x2.0 | 3 |
| | | - Waterway box below B4.0x4.0 | 2 |
| | | - Waterway box exceeding B4.0x4.0 ~ Discharge section under 30m2 | 1 |
| | | - Small bridges | 0 |



**Table 3.** Quantified rainfall criteria for *Hazard Class* of debris flow.

| Hazard class | Rainfall reoccurrenceperiod(year) | 1 hour rainfall (mm) | | 6 hour rainfall (mm) | | 3 day rainfall (mm) | |
|---|---|---|---|---|---|---|---|
| | | Accumulated rainfallrange | Rainfall Criteria | Accumulated rainfall range | Rainfall criteria | Accumulated rainfallrange | Rainfall criteria |
| S | 2~5 | 36.5~47.7 | 35 | 88.6~122.8 | 90 | 218.7~317.7 | 220 |
| A | 5~20 | 47.7~62.2 | 45 | 122.8~167.2 | 125 | 317.7~443.7 | 320 |
| B | 20~50 | 62.2~71.4 | 60 | 167.2~195.1 | 160 | 443.7~527.4 | 420 |
| C | 50~100 | 71.4~78.3 | 75 | 195.1~216.3 | 195 | 527.4~586.8 | 520 |
| D | Higher than 500 | Higher than 94.2 | 95 | Higher than 265.0 | 270 | Higher than 724.5 | 720 |



**Table 4.** State of debris flow hazard and measured rainfall of three target expressway sections.

| Expressway Section | Time of Occurrence | Rainfall (mm) | | Damage of debris flow | |
|---|---|---|---|---|---|
| | | Daily accumulated rainfall | Hourly max.rainfall | Condition | Debris sedimentation (m$^3$) |
| Pyeongchang area | 2006, 7.15., 12:00 | 244.0 | 66.0 | Road blocked | 5,000 |
| Deogyu Mountain area | 2005, 8.03., 01:00 | 312.0 | 54.5 | Road blocked | 3,000 |
| Juksan-Geochang area | 2006, 7.18., 23:00 | 121.0 | 31.5 | Drainage blocked | 1,500 |



**Table 5.** Average of *Susceptibility Value*, *Vulnerability Value*, and *Hazard Value* of occurrence and non-occurrence cases

|  | Occurrence case | | | Non-occurrence case | | |
|---|---|---|---|---|---|---|
|  | *Susceptibility value* | *Vulnerability value* | *Hazard value* | *Susceptibility value* | *Vulnerability value* | *Hazard value* |
| Pyeongchang | 11.67 | 8.56 | 20.22 | 9.36 | 7.64 | 17.00 |
| Deogyu Mountain | 11.78 | 7.44 | 19.22 | 10.14 | 7.44 | 17.57 |
| Juksan&Geochang | 11.25 | 8.33 | 19.58 | 9.63 | 8.25 | 17.88 |
| Average | 11.57 | 8.11 | 19.67 | 9.71 | 7.78 | 17.48 |





**Figure 1.** Attribute data processing based on ArcGIS Toolset.



**Figure 2.** Details of sequence application for watershed slope attributes: **(a)** polyline entity of DEM; **(b)** DEM for elevation; **(c)** DEM for slope; **(d)** flow direction; **(e)** flow accumulation; **(f)** watershed.

**Figure 3.** Details of sequence application for valley slope attributes: **(a)** elevation extracted for cells on valley path; **(b)** cell elevation converted to points; **(c)** buffer applied on valley path; **(d)** intersect of buffer and cell points.



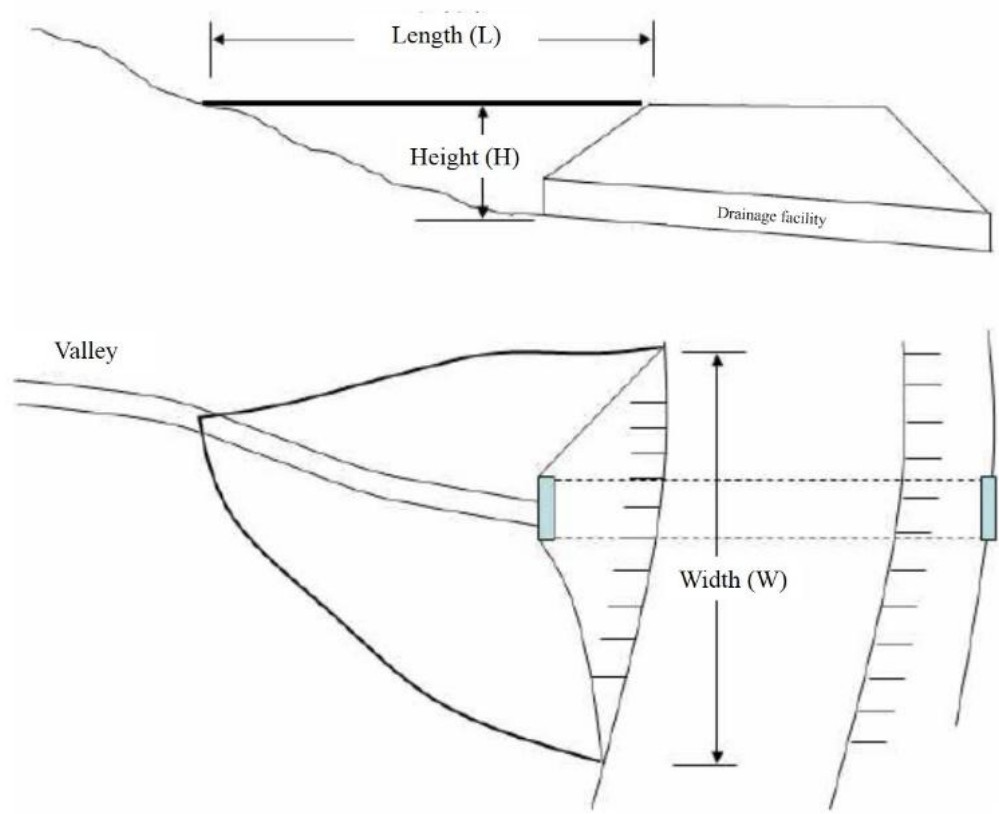

**Figure 4.** Schematic of sedimentation volume for debris flow (Expressway and Transportation Research Institute, Korea Expressway Corporation, 2009).


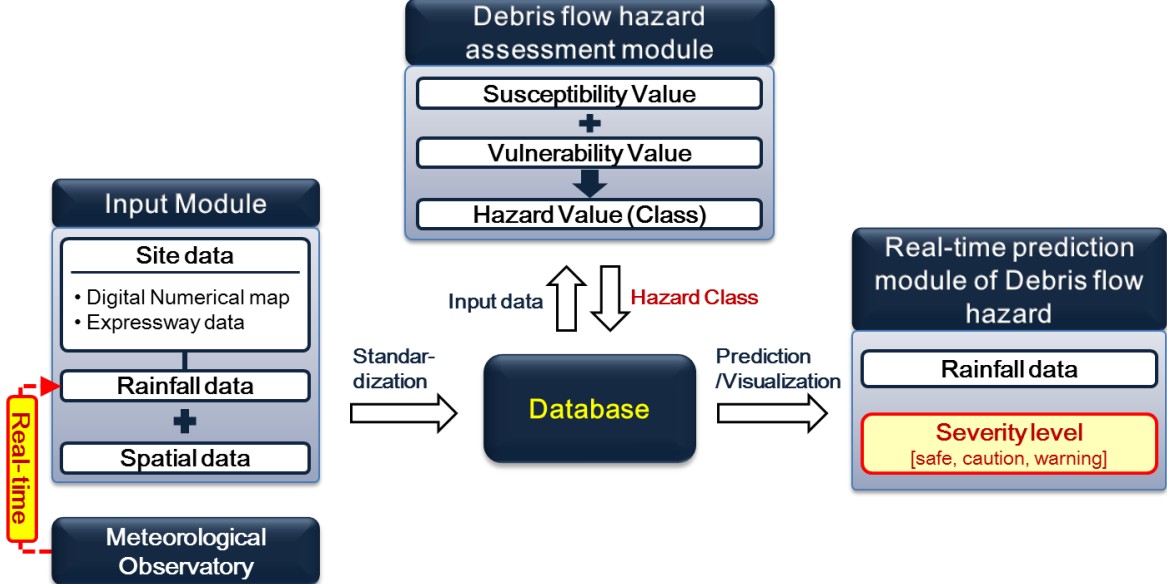

**Figure 5.** Integrated framework with debris flow hazard assessment procedures.





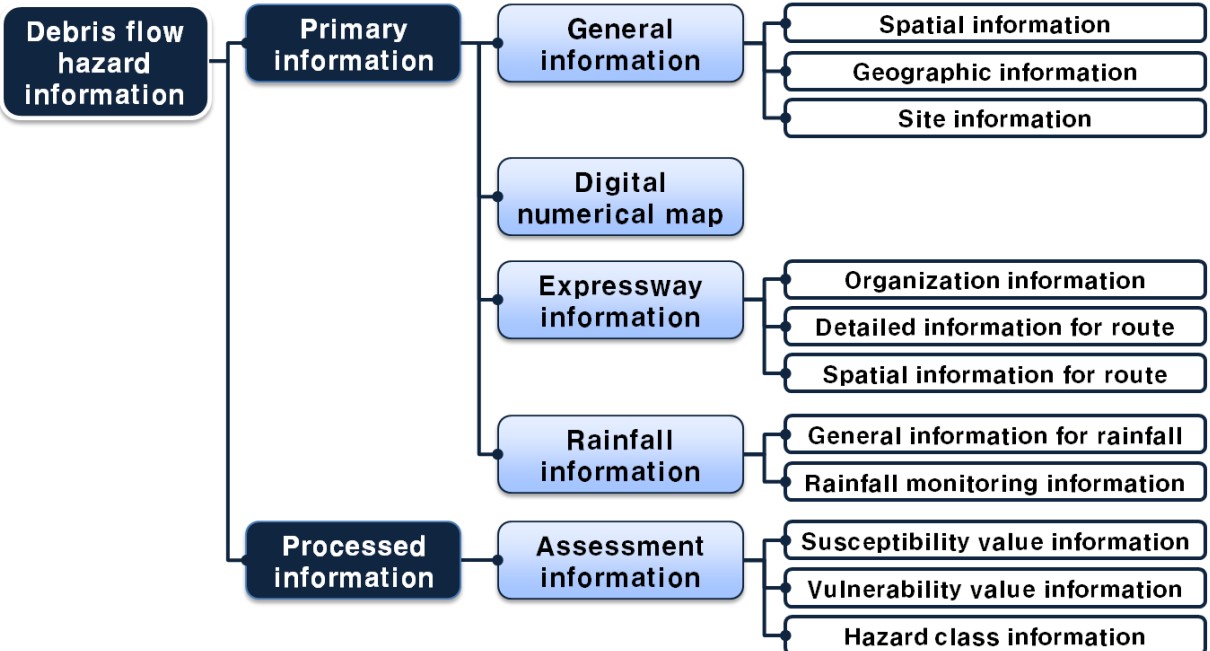

**Figure 6.** Key data classes and relations of the database for the developed system.





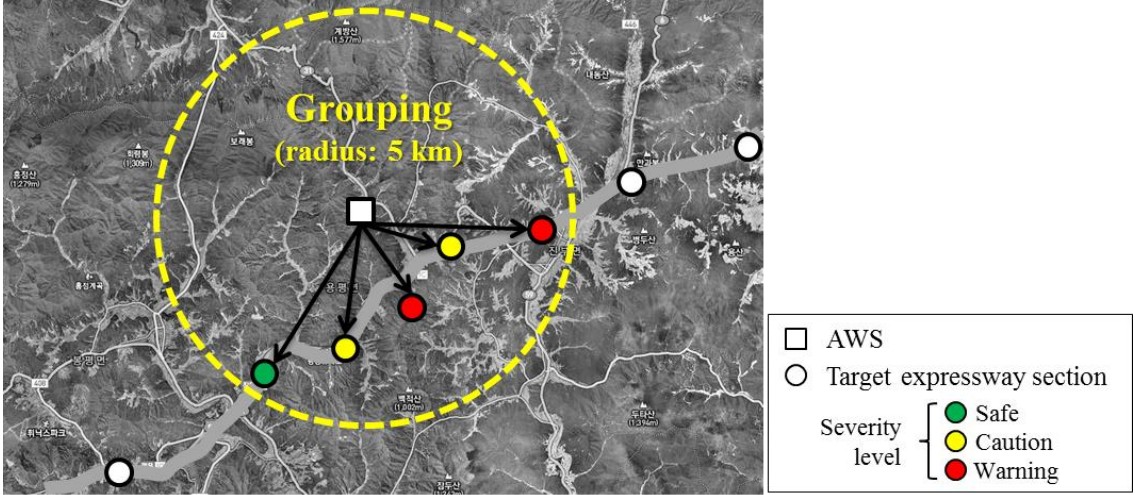

**Figure 7.** Schematic flow of transmission of rainfall data from AWS to target expressway section (point).



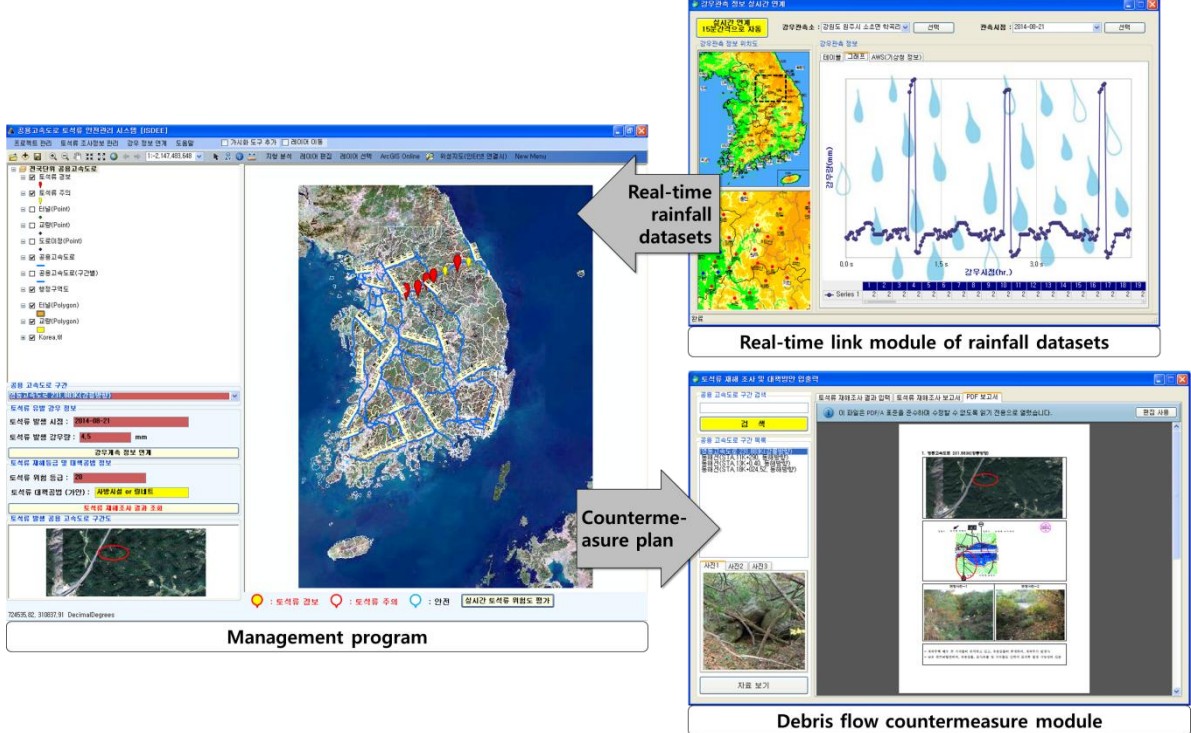

**Figure 8.** Systematic procedure of real-time debris flow hazard assessment according to system program.



**Figure 9.** Overview of assessed target regions: **(a)** Pyeongchang area of the Yeongdong Expressway; **(b)** Deogyu Mountain area of the Daejeon-Jinju Expressway; **(c)** Juksan-Geochang area of the 88 Expressway.




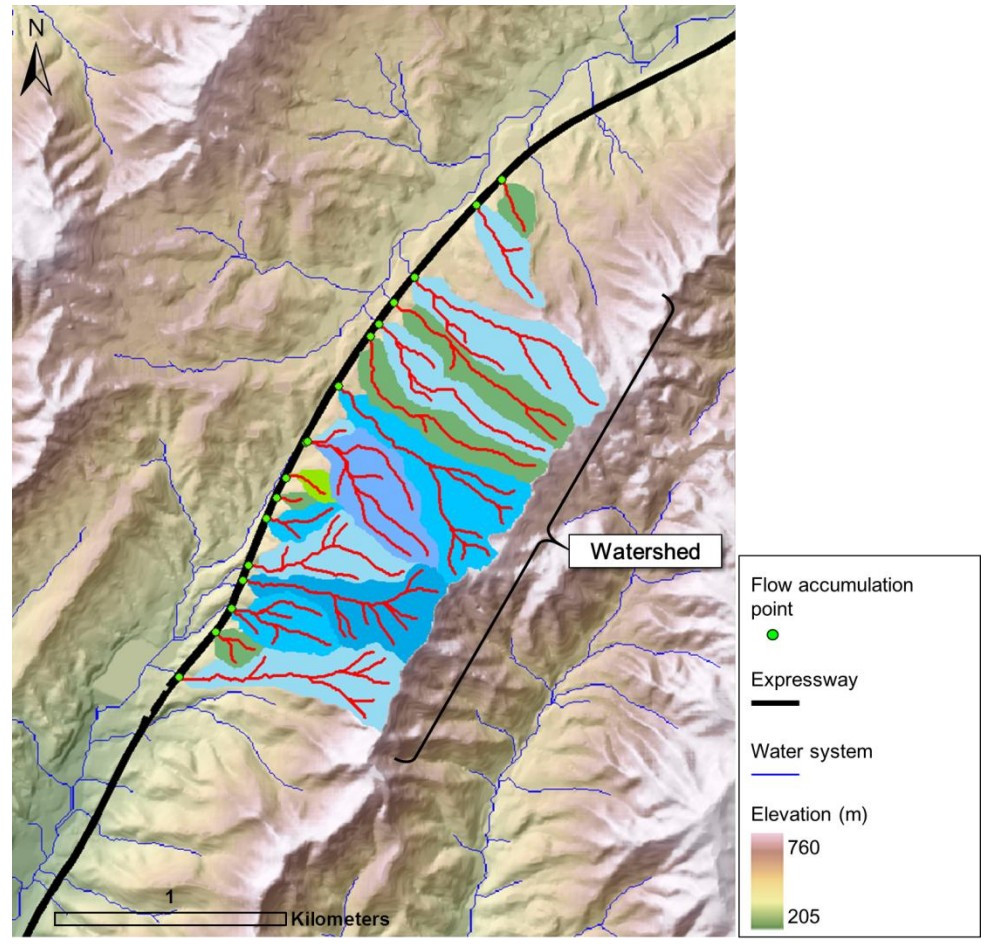

**Figure 10.** Watershed and valley slope of Jusan-Geochang area.




| | 0 | 1 | 2 | 3 | 4 | 5 | 6 | 7 | 8 | 9 | 10 |
|---|---|---|---|---|---|---|---|---|---|---|---|
| 20 | E (20) | D (21) | D (22) | C (23) | B (24) | B (25) | B (26) | A (27) | S (28) | S (29) | S (30) |
| 19 | E (19) | D (20) | D (21) | C (22) | B (23) | B (24) | B (25) | A (26) | S (27) | S (28) | S (29) |
| 18 | E (18) | D (19) | D (20) | C (21) | C (22) | B (23) | B (24) | A (25) | A (26) | △S (27) | S (28) |
| 17 | E (17) | D (18) | D (19) | C (20) | C (21) | B (22) | B (23) | A (24)■ | A (25) | S (26) | S (27)● |
| 16 | E (16) | D (17) | D (18) | C (19) | C (20) | B (21) | B (22) | ○B (23) | A (24)■ | A (25)● | S (26) |
| 15 | E (15) | E (16) | D (17) | D (18) | C (19) | C (20) | B (21) | B (22) | A (23) | A (24) | A (25)● |
| 14 | E (14) | E (15) | D (16) | D (17) | C (18) | C (19) | B (20) | B (21) | B (22)■ | A (23)● | A (24)● |
| 13 | E (13) | E (14) | D (15) | D (16) | C (17) | C (18)■ | C (19) | B (20) | B (21)■ | B (22)● | A (23) |
| 12 | E (12) | E (13) | D (14) | D (15) | C (16) | ○C (17) | C (18) | B (19) | △B (20) | B (21)▲ | B (22)● |
| 11 | E (11) | E (12) | D (13) | D (14) | C (15) | C (16) | C (17)▲ | △C (18)● | B (19) | B (20) | B (21)▲ |
| 10 | E (10) | E (11) | E (12) | D (13) | □D (14) | □C (15) | C (16) | C (17) | C (18) | ○B (19) | ○B (20) |
| 9 | E (9) | E (10) | E (11) | D (12) | D (13) | □C (14)▲ | C (15) | C (16)● | C (17)● | △C (18)● | B (19)● |
| 8 | E (8) | E (9) | E (10) | D (11) | D (12) | D (13) | C (14)■ | C (15)● | C (16) | △C (17) | □C (18)▲ |
| 7 | E (7) | E (8) | E (9) | E (10) | D (11) | ○D (12) | ○D (13) | C (14)■ | △C (15)■ | C (16) | C (17) |
| 6 | E (6) | E (7) | E (8) | E (9) | D (10) | D (11) | D (12) | D (13) | C (14) | ○C (15) | C (16) |
| 5 | E (5) | E (6) | E (7) | E (8) | E (9) | ○D (10) | D (11) | D (12) | D (13) | D (14) | D (15) |
| 4 | E (4) | E (5) | E (6) | E (7) | E (8) | E (9) | E (10) | D (11) | D (12) | △D (13) | □D (14) |
| 3 | E (3) | E (4) | E (5) | E (6) | E (7) | E (8) | E (9) | E (10) | E (11) | D (12) | D (13) |
| 2 | E (2) | E (3) | E (4) | E (5) | E (6) | E (7) | E (8) | E (9) | E (10) | E (11) | E (12) |
| 1 | E (1) | E (2) | E (3) | E (4) | E (5) | E (6) | E (7) | E (8) | E (9) | E (10) | E (11) |
| 0 | E (0) | E (1) | E (2) | E (3) | E (4) | E (5) | E (6) | E (7) | E (8) | E (9) | E (10) |
| Susceptibility/ Vulnerability | 0 | 1 | 2 | 3 | 4 | 5 | 6 | 7 | 8 | 9 | 10 |

| | Pyeongchang | Deogyu Mountain | Juksan&Geochang |
|---|---|---|---|
| Occurrence : | ● | ■ | ▲ |
| Non-occurrence : | ○ | □ | △ |

**Figure 11.** Table of *Hazard class* in accordance with *Susceptibility* and *Vulnerability Values* of debris flow occurrence and non-occurrence cases for three expressway sections.

