# Peer review of "GIS-based Real-time Framework of Debris Flow Hazard Assessment for Expressways in Korea"

_Natural Hazards and Earth System Sciences, 2016_

## Referee Comment (RC1) · Anonymous Referee #1 · 15 Feb 2016

In this paper, authors developed a method to assess debris flow hazard for express-ways in a simple way. Even though it is well described and fairly well written, the method seems unreasonable to assess the hazard level correctly. Therefore to be published in the journal, it is requried to be revised.

Basically the method results in poor assessment for the hazard based on the real occurrence data as shown in Table 5 as the authors also agreed in their paper. It seems mainly because of the scoring system that they adopted using several attributes. First of all, they included the vulnerability value in the score, but it does not distinguish between the areas of occurrence and those of non-occurrence as revealed in Table 5. Also they did not give an explanation on why they selected the attributes and gave the points according to the evenly divided values without any weightings which is very critical to review the paper.
[Figure]

I also found some typos with incorrect year compared to that in references. The references should be also reordered alphabetically.

---

## Referee Comment (RC2) · Anonymous Referee #2 · 22 Feb 2016

Authors developed a simple method (but not limited to its current one) to assess debris flow hazard for expressway management. To my knowledge, this has never been tried in Korea and the method has limitations due to the wide coverage area and usage of available data throughout the whole expressway constructed in Korea.

I agree with the first anonymous refree#1 regarding the scoring system. However, I would rather focus on the virtue of rapid decision making and wide coverage (or management system for the whole expressway system in Korea). I think the current method has permanent values that the GIS system is used for the expressway management system with the available dataset including rainfall intensity and the slope angle. The system has potential to be upgraded with more data accumulation and more case histories. I guess the scoring system itself would be updated with more case histories and dataset. I strongly recommend that the discussion on the current limitation of the

system and how the program can be developed better with more precision.

The current system might not be perfect but I, personally think that once the system is applied to the Korean expressway management system and the accumulation of data with upgrade of the scoring system, it can be a powerful tool.
* * *

---

## Referee Comment (RC3) · Anonymous Referee #3 · 24 Feb 2016

Most previous papers about GIS-based landslide assessments have been dealt generally with circular slope failures considering slope, rainfall, saturation, vegetation. This paper is about debris flow hazard assessment in a simple way. Even though the framework suggested in this paper is well constructed and operated, the adopted method to assess the hazard level is somewhat unreasonable. Therefore it is required to be revised to be published. Basically the assessment method cannot seem to consider various influence factors shown in Table 1, especially vegetation properties and geological properties. In addition, Hazard Classes as shown in Table 3 and Figure 11 are determined only based on rainfall reoccurrence period or accumulated rainfall not considering the susceptibility value and the vulnerability value. That is, the hazard value does not include hydrological properties, vegetation properties and geological properties, which are important influence factors on debris flows. Due to these lacks of

considerations in the adopted assessment method, the framework seems to result in poor assessment for the hazard as the authors also agreed in their paper.

I also found some typos like the first anonymous refree#1. Typos are at; 20th line in page 3, 7th line in page 5, 2nd line in page 6, 17th line in page 6, 24th line in page 6, 1st line in page 7, 29th line in page 8, 20th line in page 10 (not occured in Class E in Fig. 11) and in figure 2 : necessary to add the process to consider rainfalls. in table 2 : the unit of discharge section area of waterway box Figure 11 is not figure but table.

---

## Author Comment (AC1) · 22 Mar 2016

Authors really appreciate reviewer's valuable and encouraging comments on our manuscript. Also, the original manuscript was corrected and improved considering the remarks and suggestions from reviewer acting as a referee. So in this response to reviewer comments, the revision of the original manuscript and point-by-point response were prepared. Author's corrections were described in the Revisions #1 to #3 presented below. The reply for anonymous referee#1 and according revised manuscript were prepared as additional attached supplement files (zip).

Please also note the supplement to this comment:
http://www.nat-hazards-earth-syst-sci-discuss.net/nhess-2016-2/nhess-2016-2-AC1-supplement.zip

---

## Author Comment (AC3) · 22 Mar 2016

Thank you very much for referee's sincere comments. Authors corrected the pre-existing manuscript taking into account the comments. The corrected or additionally incorporated parts in the revised manuscript were underlined and red letters. For the specific comments from anonymous referee #3, authors' corrections are as follows: Authors appreciate referee's sincere review on our manuscript to making the paper near perfect and clarify the novelty of manuscript. Also, the original manuscript was corrected and improved considering the remarks and suggestions from reviewer acting as a referee. So in this response to reviewer comments, the revision of the original manuscript and point-by-point response were prepared. Author's corrections were described in the revisions #1 to #3 for comments.

The reply for anonymous referee#3 and according revised manuscript were prepared as additional attached supplement files (zip).

Please also note the supplement to this comment:
http://www.nat-hazards-earth-syst-sci-discuss.net/nhess-2016-2/nhess-2016-2-AC3-supplement.zip
* * *